# Memory-Efficient Activation Checkpointing with Sliding Window and Hirschberg's Algorithm for 0/1 Knapsack Solving in PyTorch

**Jędrzej Maczan**
Cohere Labs Community
Poland
jedrzej@maczan.pl

## Abstract

Activation checkpointing minimizes the runtime of neural networks under a given memory budget, by selecting which intermediate tensors to store and which to recompute. PyTorch solves this as a 0/1 knapsack problem, where operations from a joint forward-backward computation graph are items with a memory cost (weight) and a runtime saving (value). The default solver, `dp_knapsack`, allocates a full dynamic programming (DP) table of shape $(n + 1) \times (W + 1)$, where $n$ is the number of operations and $W$ is the quantized memory budget. This method is resource-hungry and crashes at $n = 100$ items on a machine with 64 GB RAM.

In this paper, we introduce `dp_knapsack_sliding_hirschberg`, which combines the sliding window trick and Hirschberg's algorithm to reduce peak memory from $O(nW)$ to $O(W)$ while preserving the exact optimal solution. Our experiments show successful knapsack execution at $n = 2,000$ (peak 58.4 GB), where `dp_knapsack` fails at $n = 100$, a **20$\times$ increase in computable problem size**. In addition, our benchmarks show a consistent **25–28% runtime speedup** over `dp_knapsack`.

The implementation is merged into PyTorch and released in version 2.10.

## 1 Introduction

*Activation checkpointing* [Chen et al., 2016] is a method of minimizing program runtime. PyTorch [Paszke et al., 2019] implements this in `torch.compile`, where it captures the graph [Reed et al., 2021] and uses `min_cut_rematerialization_partition` to chooses which operations to save or recompute given a budget [PyTorch Team, 2025]. Algorithmically, this is a 0/1 knapsack problem, where every candidate operation is an *item* with a memory cost (weight) and a runtime saving (value), and the goal is to maximize the total savings within a given capacity.

The default solver, `dp_knapsack`, uses standard DP with a full 2D table of shape $(n + 1) \times (W + 1)$, where $n$ is the number of operations and $W$ the quantized memory budget. We empirically show that problems with as few as $n = 100$ items result with out-of-memory (OOM) crashes on 64 GB hardware. We introduce `dp_knapsack_sliding_hirschberg`, which combines two algorithmic improvements over the default solver: (i) a **sliding window** that allows to store only two rows of the DP table, thus reduces memory usage from $O(nW)$ to $O(W)$, and (ii) **Hirschberg algorithm**, which recovers the optimal item selection from these two rows. Their application to PyTorch was suggested in a TODO comment in the original `dp_knapsack` source code. Our contribution is successful combination of sliding window and Hirschberg algorithm as a knapsack solver in PyTorch, implementation shipped in production to the machine learning community in PyTorch 2.10 and an empirical evaluation of the memory and runtime gains over the default solver.

Preprint.

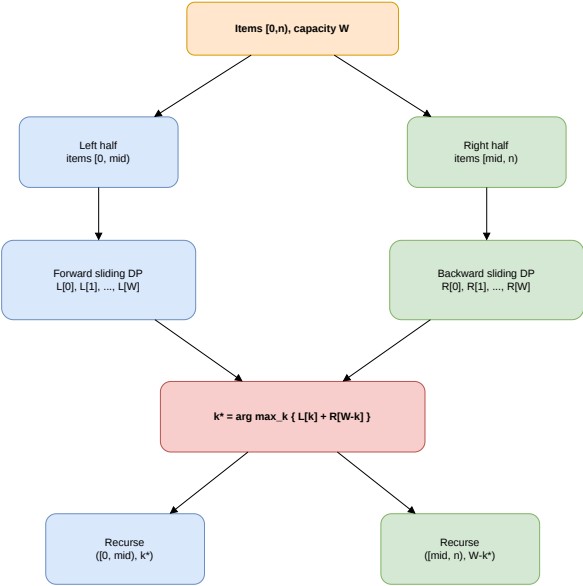

Figure 1: DP Hirschberg with sliding-window. The item set is split in the middle. Forward and backward DP passes compute vectors L and R. The optimal split $k\text{*}=\text{argmax}_k L[k] + R[W - k]$ creates two subproblems.

## 2 Background

**0/1 knapsack and dynamic programming.** Given $n$ items with weights $w_i$ and values $v_i$ and a capacity $W$, the we search for such $S \subseteq [n]$ with $\sum_{i \in S} w_i \leq W$ that they maximize $\sum_{i \in S} v_i$. A classical DP solution [Bellman et al., 1957] builds a full table $T[i][c]$ - best value using items $1, \ldots, i$ under capacity $c$ - [Kellerer et al., 2004, Martello and Toth, 1990, Pisinger, 1997] with $T[i][c] = \max(T[i - 1][c], T[i - 1][c - w_i] + v_i)$ if $w_i \leq c$. This approach takes $O(nW)$ time and space and recovering the selected items requires backtracking through all the $n$ rows. Checkpointing originates from reverse-mode automatic differentiation [Griewank, 1992, Griewank and Walther, 2000] and PyTorch's `min_cut_rematerialization_partition` automates the save/recompute decision [PyTorch Team, 2025, He and Yu, 2023]. Earlier work used DP to decide which activations to store under a budget [Gruslys et al., 2016]. Reversible layers avoid storage entirely [Gomez et al., 2017, Kitaev et al., 2020].

**Sliding window and Hirschberg's algorithm.** Since only the optimal *value* (the last column and row) is needed in practice for memory planning purposes, two rows of the DP table are enough. This change alone reduces memory use to $O(W)$ [Cormen et al., 2022]. We could end at this optimization if we didn't need the items *selection*, but in case of memory planning we obviously need it. Hirschberg [1975] introduced a divide-and-conquer algorithm that achieves linear memory use while recovering the full and exact solution, trading a constant factor of extra time [Llorens and Vilar, 2022]. This algorithm splits the items into two halves, computes each half's DP profile in $O(W)$ space using the sliding window, combines the profiles (DP rows) to find the optimal split and recurses. This was adapted to knapsack in competitive programming [Ciobanu, 2016] and is the basis for our solver.

## 3 Method

`dp_knapsack_sliding_hirschberg` combines the sliding window with Hirschberg's divide-and-conquer to solve knapsack in $O(W)$ space while recovering the exact selection. To avoid stack overflow for large $n$, it uses an explicit LIFO stack (Algorithm 1). Each frame stores an index range $[\ell, r)$ and capacity $c$. A base case ($r - \ell = 1$) saves the single item if its weight fits and its value is positive. Zero-weight items are always saved. For all the other cases, the range is split in the middle,

---

**Algorithm 1** `dp_knapsack_sliding_hirschberg`

---

**Require:** Items $(w_i, v_i)_{i=0}^{n-1}$, capacity $W$
**Ensure:** Lists *saved*, *recomputable*

1:  *saved* $\leftarrow$ [];   *recomputable* $\leftarrow$ [];   *stack* $\leftarrow$ [(0, $n$, $W$)]
2: **while** stack is not empty **do**
3:    $(\ell, r, c) \leftarrow$ *stack.pop*();   **if** $r - \ell = 0$ **then continue**
4:    **if** $r - \ell = 1$ **then**
5:       **if** $w_\ell \leq c$ **and** $v_\ell > 0$ **then** *saved.append*($\ell$) **else** *recomputable.append*($\ell$)
6:    **else**
7:       $m \leftarrow \ell + \lfloor (r - \ell)/2 \rfloor$
8:       $P_1 \leftarrow$ SLIDINGWINDOWDP($\ell, m, c$);    $P_2 \leftarrow$ SLIDINGWINDOWDP($m, r, c$)
9:       $c^* \leftarrow \arg\max_{0 \leq k \leq c} (P_1[k] + P_2[c - k])$          $\triangleright$ $P_2$ accessed in reverse
10:     *stack.push*(($m$, $r$, $c - c^*$));   *stack.push*(($\ell$, $m$, $c^*$))
11:    **end if**
12: **end while**
13: **return** SORT(*saved*),  SORT(*recomputable*)

---

a sliding-window DP profile is computed for each half, and the argmax of the combined profiles gives the optimal capacity split $c^*$ (Figure 1)

SLIDINGWINDOWDP($\ell, r, c$) computes the DP profile for items $\ell, \ldots, r - 1$ within capacity $c$, using two row buffers of size $c + 1$.

**Space:** four DP buffers of size $W + 1$, a stack of $O(\log n)$ frames, and output lists of size $n$, giving $O(W + n)$ peak space versus $O(nW)$ for `dp_knapsack`.

**Time:** each of $O(\log n)$ levels does $O(nW)$ work, so $O(nW \log n)$ versus $O(nW)$ (see Section 4).

## 4 Evaluation

We benchmark all four PyTorch knapsack solvers on synthetic instances mirroring real activation memory planning ($W$ from $1.4 \times 10^8$ at $n$=1 to $3.8 \times 10^8$ at $n$=100), each averaged over 1,000 runs on an idle machine (Ubuntu 24.04.2, AMD Ryzen 7 9800X3D, 64 GB RAM). Table 1 and Figure 2 show runtime and exactness. `dp_knapsack_sliding_hirschberg` is consistently 25–28% faster than PyTorch's default `dp_knapsack` (Figure 2, left), `ilp_knapsack` is faster still but requires SciPy dependency, and `greedy_knapsack` is fastest but suboptimal.

`dp_knapsack` allocates a $(n+1) \times (W+1)$ table: at $n$=100, $W \approx 3.8 \times 10^8$ this is $\approx 304$ GB, and it fails with an out-of-memory error on a 64 GB machine. `dp_knapsack_sliding_hirschberg` uses a fixed number of row buffers and runs at $n = 2{,}000$ with a 58.4 GB peak on the same machine, resulting with a **20× increase in computable problem size**, with the DP table shrinking from $\sim$304 GB to $\sim$6 GB at $n$=100.

The three solvers (`dp_knapsack`, `ilp_knapsack`, `dp_knapsack_sliding_hirschberg`) produce the exact, optimal solution at every size, while `greedy_knapsack` produces up to 7.4% below optimal (Figure 2, right). This difference between `greedy_knapsack` and `dp_knapsack_sliding_hirschberg` makes the latter a substantial improvement to PyTorch ecosystem - it keeps `dp_knapsack`'s exactness while removing its memory bottleneck. Thus, as a rule of thumb we recommend `ilp_knapsack` when SciPy is available, `ilp_knapsack` when exact solutions don't matter and `dp_knapsack_sliding_hirschberg` otherwise for large or memory-constrained graphs.

## 5 Discussion and Related Work

`dp_knapsack` fails when $n$ is large - with long sequence lengths, wide graphs, or full model graph captures in `torch.compile` [Shoeybi et al., 2019, Narayanan et al., 2021]. `dp_knapsack_sliding_hirschberg` has worse asymptotic time than `dp_knapsack` ($O(nW \log n)$ vs. $O(nW)$) [Llorens and Vilar, 2022]. Yet it comes with runtime speedup, thanks to

Table 1: Mean solver runtime in seconds over 1,000 runs

| Solver | $n = 5$ | $n = 10$ | $n = 20$ | $n = 50$ | $n = 100$ |
|---|---|---|---|---|---|
| `greedy_knapsack` | $1.1 \times 10^{-6}$ | $1.4 \times 10^{-6}$ | $2.2 \times 10^{-6}$ | $4.8 \times 10^{-6}$ | $8.9 \times 10^{-6}$ |
| `ilp_knapsack` | $5.0 \times 10^{-4}$ | $9.6 \times 10^{-4}$ | $6.6 \times 10^{-4}$ | $9.7 \times 10^{-4}$ | $1.3 \times 10^{-2}$ |
| `dp_knapsack` | 0.107 | 0.602 | 1.716 | 11.613 | OOM |
| OURS | 0.079 | 0.473 | 1.244 | 8.396 | 35.10 |

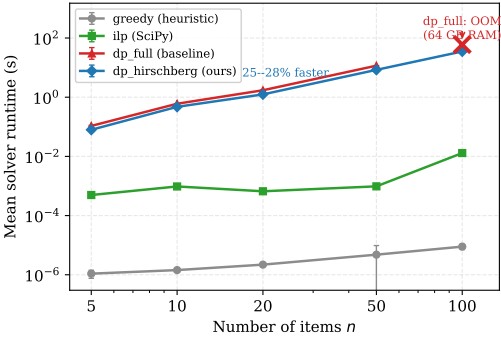 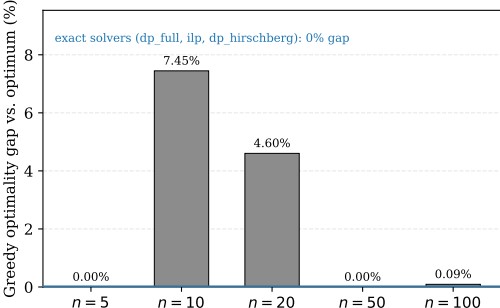

Figure 2: **Left:** mean solver runtime versus $n$. **Right:** `greedy_knapsack` leaves an instance-dependent optimality gap of up to 7.4%, whereas the exact solvers (`dp_knapsack`, `ilp_knapsack`, `dp_knapsack_sliding_hirschberg`) all reach the optimum.

using fixed buffers, and optimizing memory-access patterns, which are shown to dominate performance [Dao et al., 2022]. The quality, usefulness in the mainstream deep learning framework and correctness of `dp_knapsack_sliding_hirschberg` was reviewed and merged into PyTorch and shipped in PyTorch 2.10 [PyTorch Team, 2026].

Activation checkpointing [Chen et al., 2016] is now considered as a standard technique in machine learning compilers. Subsequent work explores rematerialization strategies [Kirisame et al., 2020, Beaumont et al., 2021, Schuler et al., 2022, Kumar et al., 2019, Kusumoto et al., 2019], optimal graph partitioning [Herrmann et al., 2019], and automated systems [Jain et al., 2020]. These are complemented by techniques like optimizer-state sharding, offloading [Rajbhandari et al., 2020, Rasley et al., 2020, Ren et al., 2021] and selective recomputation [Korthikanti et al., 2022], and are catalogued in broader surveys [Tian et al., 2026]. The sliding-window reduction is a standard textbook technique [Cormen et al., 2022, Kellerer et al., 2004], and Hirschberg's algorithm [Hirschberg, 1975] was originally proposed for sequence alignment.

# 6 Conclusion

We presented `dp_knapsack_sliding_hirschberg`, a memory-efficient exact 0/1 knapsack solver for PyTorch's activation memory planning. It combines the sliding window with Hirschberg's divide-and-conquer algorithm and reduces peak solver memory from $O(nW)$ to $O(W)$, while preserving the optimal solution. It empirically shows up to $20\times$ increase in computable problem size and a 25–28% runtime speedup compared to PyTorch default knapsack solver. `dp_knapsack_sliding_hirschberg` is merged into PyTorch since 2.10.

**Broader Impact Statement**

This work improves the memory efficiency of an internal solver used during PyTorch compilation, allowing to run larger models. As a general-purpose optimization that does not change model behavior, we foresee no specific negative societal impacts.

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
