# OpenReview forum: "Memory-Efficient Activation Checkpointing with Sliding Window and Hirschberg's Algorithm for 0/1 Knapsack Solving in PyTorch"
_FastML/2026/Conference — FastML 2026 Conference Submission_

### Official Review · Reviewer_iDY8 · 2026-07-18
**Memory efficient knapsack solver for PyTorch checkpointing.**

**Rating:** 3
**Confidence:** 4

**Review:**

This paper addresses a genuine bottleneck and proposes dp_knapsack_sliding_hirschberg, a memory-efficient exact solver for the 0/1 knapsack formulation used in PyTorch's activation checkpointing. By combining the classic sliding-window DP memory reduction with Hirschberg's divide-and-conquer algorithm, the authors reduce peak memory from O(nW) to O(W) while preserving exact optimality, at the cost of an O(log n) time factor. The idea itself is not new (the paper states the combination was suggested in a TODO comment in PyTorch's own source and previously discussed in a competitive-programming forum post), but its concrete implementation and integration into a widely used framework is the paper's practical contribution.

Quality: The quality of this engineering-focused paper is exceptionally high. The author identifies a critical failure point in PyTorch's activation checkpointing: out-of-memory (OOM) crashes at n=100 operations on 64 GB hardware due to O(nW) dynamic programming table allocation. The proposed dp_knapsack_sliding_hirschberg elegantly resolves this using a sliding window and Hirschberg’s divide-and-conquer algorithm. The empirical validation is rigorous, benchmarking against greedy, integer linear programming (ILP), and baseline solvers. The integration into PyTorch 2.10 confirms its robustness.

Clarity: The submission is clear and logically structured. The background of the 0/1 knapsack problem in memory planning is articulated cleanly. Algorithm 1 and Figure 1 provide a precise blueprint of the system's integration. The results are communicated efficiently without unnecessary jargon, remaining comprehensible to the interdisciplinary fast machine learning community.

Originality: While sliding window and Hirschberg’s algorithms are standard textbook concepts, the originality lies in their applied systems engineering. Tackling a recognized implementation hurdle, the author successfully combines these techniques to yield an exact knapsack solver that functions within the constraints of a major deep learning compiler.

Significance: The significance for scalable machine learning is substantial. Shrinking the memory footprint from O(nW) to O(W) drops peak memory at n=100 from roughly 304 GB to 6 GB. This expands the computable problem size by 20x. Furthermore, optimized memory access patterns yield a 25-28% runtime speedup, allowing researchers to compile significantly larger scientific models.

Pros
- Resource Reduction: Eliminates the OOM bottleneck by reducing peak memory to O(W).
- Runtime Speedup: Achieves a 25-28% runtime improvement via cache-localized memory access.
- Practical Impact: Production-ready and merged into PyTorch 2.10.
- Mathematical Exactness: Preserves the optimal solution, unlike greedy heuristics.

Drawbacks
- No End-to-End Evaluation: Lacks training benchmarks to demonstrate the net impact on real-world scientific architectures.
- Unexplored Overhead: Asymptotic time increases to $O(nW \log n)$ without an analysis of when this penalty outweighs cache benefits.